# Two-Scale Latent Dynamics for Recurrent-Depth Transformers

Francesco Pappone[1,2], Donato Crisostomi[1], and Emanuele Rodolà[1]

[1]Sapienza University of Rome
[2]PSTP Technoscience

## Abstract

Recurrent-depth transformers scale test-time compute by iterating latent computations before emitting tokens. We study the geometry of these iterates and argue for a simple, *two-scale* operational picture: (i) within a looped block, updates act as *small-scale refinements*; (ii) across consecutive blocks, states undergo a *larger-scale drift*. Across training, our measurements show that loop steps become *smaller* and increasingly *orthogonal* to one another, indicating better local modeling of fine structure rather than merely pushing in a single direction. These dynamics motivate an early-exit mechanism based on the model's second-order difference in step-size, which we show is superior in terms of performance, stability and time-efficiency, when compared to the KL-divergence exit strategy of Geiping et al. [5] and its naive first-order counterpart.

## 1 Introduction

How should a language model spend its test-time compute (TTC)? One option is to generate more tokens, externalizing "thinking" into text as in recent chain-of-thought and deliberation systems [11, 1], but this quickly inflates latency and context length. A complementary option is to think *in latent space*: iterate internal computations before emitting a token so that extra FLOPs improve the representation rather than relying on increasing the sequence length. Recent work shows that looping a Transformer block in this way can yield better reasoning at inference-selected iteration counts [5, 6].

In this paper, we ask the simple question: *what geometry emerges when a block is looped?* Our thesis is that recurrent depth naturally separates representations across two scales. Within a looped block the model performs *local refinements*: updates become small and increasingly orthogonal, tracing a stable-curvature spiral around a nearby representation. Across blocks the representation undergoes a *slower drift*, reflecting coarser changes introduced by depth.

We support this with iterate-level measurements, step norms and consecutive-step angles, tracked over training and visualized with PCA trajectories. Across three recurrent regions in a GPT-2–like model, we consistently observe (i) rapid decay of step size within 5–10 loop steps and (ii) stabilization of the angle statistic at moderate values, indicating non-collinear refinements rather than repeated pushes in one direction. PCA plots echo this picture: tight arcs inside loops, larger jumps at block hand-offs.

**This paper.** We take a geometry-first look at looped blocks. We show that as training proceeds: (i) loop steps shrink quickly (small-scale refinements), and (ii) consecutive steps become more orthogonal (updates sweep around rather than continue straight). Across blocks, representations change more coarsely (slow drift). Motivated by this pattern, we propose a simple, *second-order* early-exit rule based on the norm of the difference between consecutive loop updates ("acceleration").

Preprint.

Unlike norms or KL on decoded distributions, acceleration triggers precisely when the local spiral stabilizes, enabling earlier, quality-preserving exits.

Wrapping up, our contributions are the following:

- A group-wise extension of Geiping et al. [5]'s recurrent-depth approach, with recurrence happening independently in different, disjoint sets of blocks in the transformer.
- A diagnostic for loop geometry (step norms and consecutive-step angles) showing shrinking updates and increasing orthogonality over training.
- Evidence, across three recurrent regions of a GPT-2–like model, that looped blocks perform fine-grained refinements while cross-block changes produce slower drift.
- A *second-order* exit criterion that halts when consecutive updates stop changing, delivering a better latency-quality trade-off than step-norm and KL-based baselines.

**Relation to Geiping et al. (2025)**

Geiping et al. [5] loop recurrent blocks at test time and study halting via step-norm and KL on decoded distributions, also noting spiral-like iterate behavior in 2D PCA. Our contribution is complementary and more granular: (i) we frame a *two-scale* geometry (small, increasingly orthogonal refinements inside loops; coarser cross-block drift), (ii) we *quantify* this with iterate-space diagnostics (step norms, consecutive-step angles) *and* cross-block drift metrics, and (iii) we introduce a *decoding-free, second-order* halting rule ("acceleration") with a two-hit check. Unlike KL-based exits, our criterion has $\mathcal{O}(d)$ per-token-step cost and requires no logits decoding; unlike step-norm, it is sensitive to local *curvature/rotation* and avoids premature halts under stable-speed spirals.

## 2 Experimental setup

**Backbone and data.** We train a GPT-2-style decoder-only transformer [13, 10] (12 layers, 12 heads, hidden size 768, block size 512) using the same web-text (Fineweb [12]) pretraining setup for all experiments. When initializing parts of the model that are applied recurrently (recurrent regions) we use the looped-input projection technique of Geiping et al. [5]. For independently recurring blocks, we inject random noise each time the block is applied (combined with the input sequence) mirroring Geiping et al.'s method for a single repeating block, but extended here to each block group.

**Recurrence pattern.** We enable recurrence in three **groups**: layer **4** (self-loop), layers **5–6** (paired loop), and layer **7** (self-loop). All other layers run single pass.

We collect loop trajectories on held-out text. For tokens and loop step $k$ in group $g$, with hidden $x_g^{(k)} \in \mathbb{R}^d$, we define

$$\Delta_g^{(k)} := x_g^{(k+1)} - x_g^{(k)} \,. \tag{1}$$

We track (i) **step size** $\|\Delta_g^{(k)}\|_2$ and (ii) **step orthogonality** $\cos\angle\big(\Delta_g^{(k)}, \Delta_g^{(k-1)}\big)$. Statistics are averaged across tokens with $1\sigma$ bands.

**What we measure.** We focus on two model-agnostic, iterate-only diagnostics: *radial refinement*, the rapid decay of $\|\Delta^{(k)}\|_2$ within a small number of loop steps; and *angular refinement*, the stabilization of $\cos\angle\big(\Delta^{(k)}, \Delta^{(k-1)}\big)$ indicating non-collinear, refinement-like updates.

## 3 Empirical observations and loop dynamics

We observe three consistent patterns across block groups (4, 5–6, 7) and checkpoints: (i) loop-step sizes shrink rapidly (the mean $\|\Delta^{(k)}\|_2$ decays quickly with $k$, typically within 5–10 steps), which is consistent with small, diminishing refinements; (ii) consecutive updates become more orthogonal: $\cos\angle\big(\Delta^{(k)}, \Delta^{(k-1)}\big)$ rises from noisy/low values and settles at lower levels ($\approx 0.5$–$0.65$ in later checkpoints), indicating complementary rather than collinear pushes; and (iii) this separation strengthens over training, with faster norm decay and steadier angles in later checkpoints, while across-block changes remain comparatively larger (slow drift).

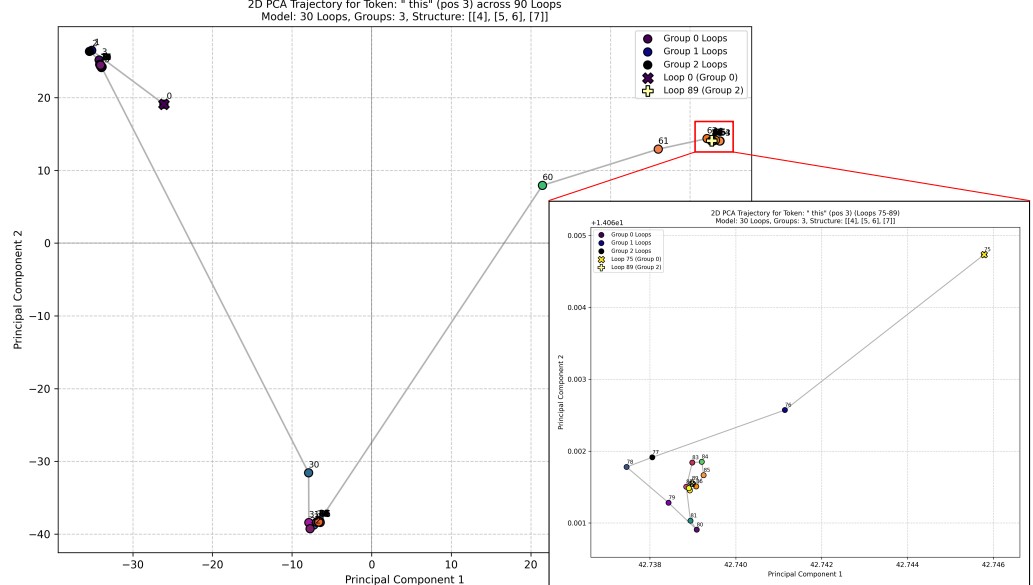

Figure 1: **Overall trajectory with inset zoom** (2D PCA). Tight loop refinements (inset) vs. larger cross-block moves (main view).

**Supporting visualization: PCA trajectories.** To visualize geometry directly, we project hidden states to 2D via PCA fit on the union of all states for the shown sequence (across depth and loop steps), as in [5]. Each polyline follows a token's trajectory in projected space; colors encode iteration order (lighter→darker). For each sequence, we refit PCA on the combined set of states so that movements along the loop and depth dimensions are directly comparable. We observe a consistent pattern: tight arcs appear within each loop, while larger jumps mark the transitions between blocks.

### 3.1 Cross-block drift

Let $x_g^{(k)}$ be the hidden state at loop step $k$ within group $g$, and $\Delta_g^{(k)} = x_g^{(k+1)} - x_g^{(k)}$. We introduce a *Drift-to-Loop Ratio* (DLR) to quantitatively characterize the two-scale dynamics behavior:

**Drift-to-Loop Ratio (DLR).** For a boundary $g \rightarrow g+1$,

$$\text{DLR}_{g \rightarrow g+1} = \frac{\left\| x_{g+1}^{(0)} - x_g^{(K_g)} \right\|_2}{\frac{1}{K_g} \sum_{k=0}^{K_g-1} \left\| \Delta_g^{(k)} \right\|_2}, \tag{2}$$

where $K_g$ is the number of loop steps used in group $g$. Larger DLR indicates a coarser cross-block move relative to within-loop refinements.

Across checkpoints, we observe (Fig. 2), DLR $\gg 1$ at block hand-offs, corroborating a *coarse* cross-block drift atop *fine* loop refinements.

## 4 Geometry-inspired early exit and trade-offs

We evaluate three exit rules and highlight their computational cost and typical failure modes. In particular:

1. **Step-norm (Geiping et al. [5])**: trigger when the update magnitude is small,

$$\|\Delta^{(k)}\|_2 < \tau. \tag{3}$$

*Pros:* cheap ($\mathcal{O}(d)$ per token-step), no decoding. *Cons:* sensitive to feature scaling; can *mis-halt* under spiral dynamics where speed plateaus but direction keeps changing. A normalized variant $\tilde{s}^{(k)} = \|\Delta^{(k)}\|_2/(\|x^{(k)}\|_2 + \varepsilon)$ reduces scale sensitivity.

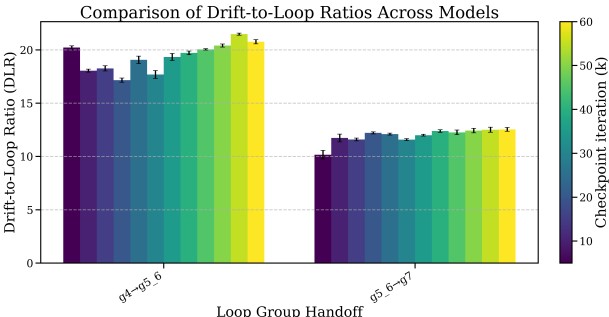

Figure 2: **Cross-block drift (DLR).** DLR at boundaries $(4 \to 5\text{–}6)$ and $(5\text{–}6 \to 7)$ across checkpoints. Values $> 1$ indicate larger-scale drift across blocks.

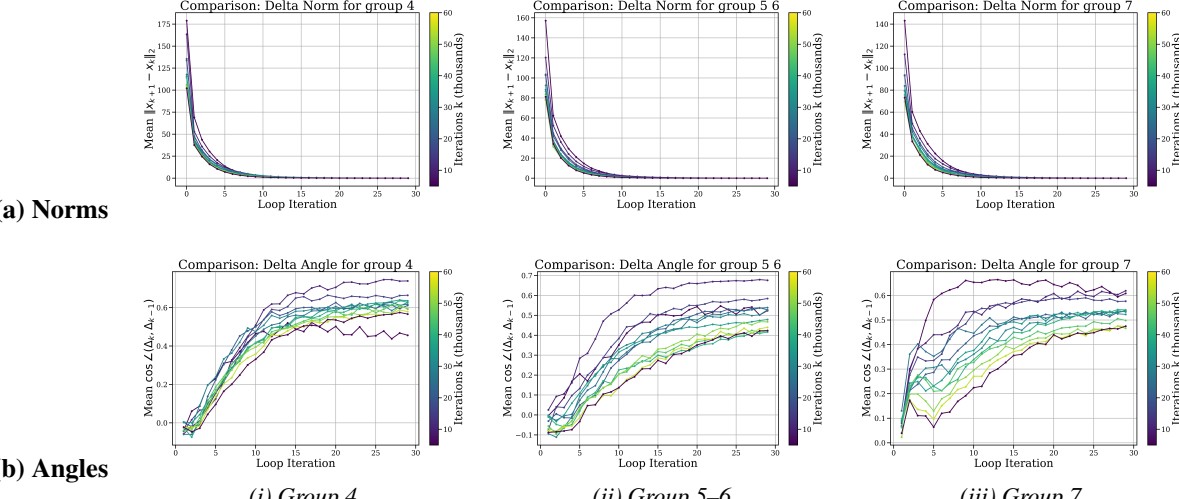

**(a) Norms**

**(b) Angles**

*(i) Group 4*         *(ii) Group 5–6*         *(iii) Group 7*

Figure 3: **Loop dynamics.** Rows show (a) step norms and (b) step angles. Within each row, panels (i)–(iii) correspond to groups 4, 5–6, and 7, respectively.

2. **KL-divergence (Geiping et al. [5])**: trigger when decoded distributions stabilize,

$$\mathrm{KL}\Big(p^{(k)}\Big\|p^{(k-1)}\Big) < \tau, \quad p^{(k)} = \mathrm{softmax}(z^{(k)}). \tag{4}$$

*Pros:* tied to the model's predictive distribution; largely invariant to latent rescalings. *Cons:* requires decoding over the vocabulary ($\mathcal{O}(V)$); depends on calibration/temperature; reacts later when small latent rotations still change logits. The approach is stable across a wide $\tau$ range but typically slower due to the softmax/KL computation requiring the use of the model's decoder head.

3. **Acceleration (ours)**: detect when the *change of the update* becomes small,

$$a^{(k)} := \big\|\Delta^{(k)} - \Delta^{(k-1)}\big\|_2, \tag{5}$$

and exit when $a^{(k)} < \tau$ *for two consecutive steps* (requires $k \geq 2$). *Pros:* cheap like step-norm ($\mathcal{O}(d)$); directly sensitive to *curvature/rotation*. *Cons:* still affected by global rescaling of latents (mitigated by a normalized form). We use a *two-hit* check to avoid spurious single-step dips; a bounded, normalized form

$$\hat{a}^{(k)} = \frac{\big\|\Delta^{(k)} - \Delta^{(k-1)}\big\|_2}{\big\|\Delta^{(k)}\big\|_2 + \big\|\Delta^{(k-1)}\big\|_2 + \varepsilon} \in [0, 2] \tag{6}$$

is a drop-in replacement when per-block scaling differs. Note that the denominator may also be based on the norm of activations, rather than step differences.

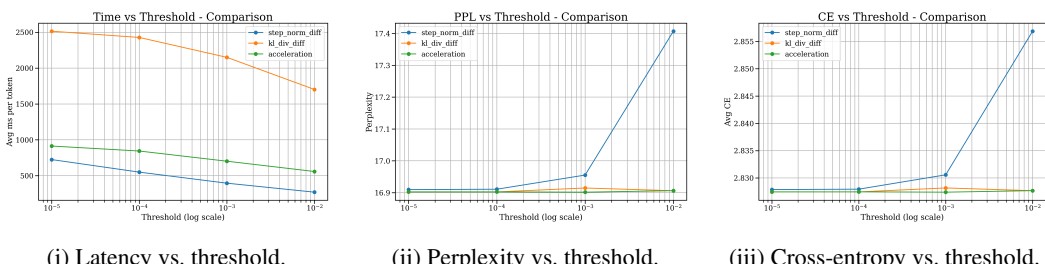

| (i) Latency vs. threshold. | (ii) Perplexity vs. threshold. | (iii) Cross-entropy vs. threshold. |

Figure 4: **Exit policies**: step-norm (blue), KL (orange), acceleration (green). Acceleration preserves quality while enabling more aggressive thresholds and lower latency; KL preserves quality but is slower; step-norm is fast but loses quality at high $\tau$.

We showed in Section 3 that looped blocks learn to make *small, increasingly orthogonal* steps, effectively tracing a spiral with nearly constant curvature. A pure norm threshold can be fooled by this regime (norms plateau at a small nonzero value due to rotation), and KL responds later because it computes and measures the decoded distribution, require the use of the decoding head of the transformer. The second-order signal $a^{(k)}$ drops precisely when the local curvature stabilizes; a two-hit check avoids spurious exits without smoothing.

**Empirical comparison.** Figure 4 summarizes the trade-offs. We see that (i) acceleration achieves large gains as $\tau$ increases (e.g., $\sim 580 \rightarrow 360$ ms/token from $10^{-5}$ to $10^{-2}$), while KL remains comparatively slow. Further, (ii–iii) acceleration and KL keep PPL/CE essentially flat across thresholds, while step-norm degrades beyond $10^{-3}$ and sharply at $10^{-2}$. These tests show that, at same quality, acceleration dominates KL on latency and avoids the step-norm quality cliff.

To summarize, the geometry of the loop dynamics naturally motivates a second-order exit criterion. Empirically, this approach achieves a better latency-quality than KL-based methods and avoids the instability of step-norm under aggressive thresholds. As a result, the method is more robust to threshold variation, reducing or even eliminating the need for extensive hyperparameter tuning.

## 5 Conclusions and limitations

We provided a geometry-first view of recurrent depth: within looped blocks, updates shrink and stabilize (small, curvature-consistent refinements), while across blocks, representations drift more gradually. Building on this, we introduced a decoding-free, second-order exit that halts when consecutive updates stop accelerating. At matched quality, it lowers latency compared to the KL-based exit of Geiping et al. [5] and avoids the instability of simple step-norm thresholds, while remaining computationally cheap.

**Limitations.** Our evidence is observational and iteration-based rather than mechanistic; PCA reduces geometry to 2D; experiments use a single GPT-2-scale model and a specific recurrence pattern; and measurements are averaged over tokens. Despite these caveats, the takeaway is simple and actionable: looped blocks handle local refinements, depth drives slow drift, and curvature stabilization offers a reliable stop signal. Future work includes scaling to larger models and datasets, learning per-block calibration for the exit rule, and integrating geometry-aware controllers with dynamic-depth or token-level TTC strategies.

## Acknowledgments and Disclosure of Funding

This work is partly supported by the MUR FIS2 grant n. FIS-2023-00942 "NEXUS" (CUP B53C25001030001), and by Sapienza University of Rome via the Seed of ERC grant "MINT.AI" (CUP B83C25001040001).

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

# Appendix

## A   Related work

Large language models deliver strong performance but incur high inference cost. One route scales test-time compute *in token space* by letting models generate longer chains of thought or deliberations, as in OpenAI's o1 and recent RL-for-reasoning works [11, 1]. This can improve quality but spends computation on extra tokens and quickly saturates the context window.

A complementary direction is to allocate compute *in latent space*. Classic recurrent computation predates Transformers [8], while within the Transformer family there are two broad threads. (i) **Dynamic depth & early exit.** Universal Transformers share parameters across depth and learn to halt (often via ACT) [2]; depth-adaptive and early-exit variants similarly modulate per-token compute by stopping when intermediate predictions stabilize (e.g., entropy/confidence heuristics) [3, 14]. (ii) **Recurrent depth in latent space.** Iterating a recurrent block to "think longer" without emitting tokens improves reasoning while letting inference decide the number of inner steps [5, 16]. Related ideas explore learned latent trajectories and continuous latent-space reasoning [4, 7], and hierarchical controllers that organize multi-stage computation [15, 9].

## B   Architecture and training details

**Backbone.**  We use a GPT-2–style decoder-only transformer [13] implemented in a minimalist NanoGPT setup [10] with learned positional embeddings (no ALiBi/rotary), 12 layers, 12 heads, hidden size $d{=}768$, and block size 512. Nonlinearities are **SiLU**. All experiments use the same **FineWeb** pretraining mixture (identical budgets across runs). We train up to 60 thousand iterations (a total of 300M tokens).

**Recurrent regions and loop sampling.**  Three depth regions are recurrent: layer **4** (self-loop), layers **5–6** (paired loop, stepped jointly), and layer **7** (self-loop). During *training*, for each recurrent group $g$ we *sample the number of loop iterations $L_g$ per forward pass* from the **lognormal-Poisson** schedule defined by Geiping et al. [5], using rate parameter $r{=}12$. For the paired group (5–6) a single draw governs both layers to preserve coordination. During *evaluation*, we either fix $L_g$ to a budget or use the early-exit criteria in the main text.

**Initialization and recurrent input projection.**  We adopt the **custom initialization** for recurrent application described by Geiping et al. [5], and their **looped-input projection** when re-feeding hidden states to a looping block. For independently recurring blocks we also inject fresh noise when feeding the recurrent input (as in Geiping et al. [5] for a single repeating block), extended here to each block group.

**Training recipe.**  Unless noted otherwise, optimizer, schedules, and regularization follow our NanoGPT baseline [10]. The only differences are: (i) custom initialization, (ii) SiLU activations, (iii) recurrent loop sampling via lognormal-Poisson ($r{=}12$), (iv) the recurrent initialization/projection above and (v) sandwich normalization (using RMSNorm).

*Summary.* Model: GPT-2–like (12L, 12H, $d{=}768$), learned PEs, SiLU, FineWeb; recurrence at 4, 5–6, 7; loop counts $L_g \sim \text{LogNormalPoisson}(r{=}12)$ per group per pass; recurrent init & projections as in Geiping et al. [5].

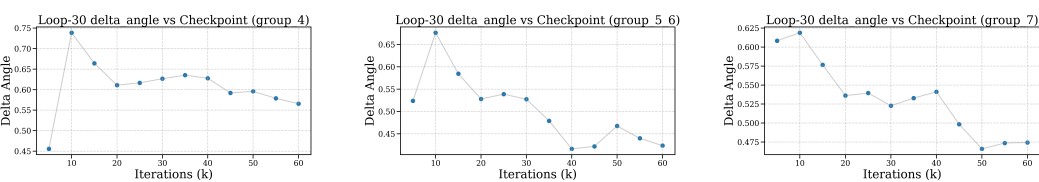

Figure 5: Fixed 30-loops $\cos \angle(\Delta^{(k)}, \Delta^{(k-1)})$ across checkpoints for groups 4, 5–6, 7.

## C  Additional plots

### C.1  Full $\Delta$ angle curves (all checkpoints)

Figure 5 reports the *full* consecutive-step cosine $\cos \angle(\Delta^{(k)}, \Delta^{(k-1)})$ for groups 4, 5–6, and 7 across all checkpoints.

## D  Early-exit implementation details

We sweep $\tau \in \{10^{-5}, 10^{-4}, 10^{-3}, 10^{-2}\}$ (log scale). Latency is average ms/token. Quality is perplexity (PPL) and cross-entropy (CE) on held-out text. The model and recurrence pattern match §2. Each point averages identical prompts across exits.

---

**Algorithm 1** Acceleration-Based Two-Hit Early Exit for Recurrent Depth

---

**Require:** Block map $f$, initial hidden state $x_0$, threshold $\tau > 0$, max steps $k_{\max}$
**Ensure:** Final state $x^\star$, steps used $k$
1: $k \leftarrow 0$;   $\delta_{\text{prev}} \leftarrow$ None;   prev_small $\leftarrow$ **false**
2: **while** $k < k_{\max}$ **do**
3:     $x_1 \leftarrow f(x_0)$ {one loop step}
4:     $\delta_{\text{cur}} \leftarrow x_1 - x_0$ {current update}
5:     **if** $\delta_{\text{prev}} \neq$ None **then**
6:        $a \leftarrow \|\delta_{\text{cur}} - \delta_{\text{prev}}\|_2$ {acceleration}
7:        small $\leftarrow (a < \tau)$
8:        **if** small **and** prev_small **then**
9:           **break** {two-hit exit}
10:       **end if**
11:       prev_small $\leftarrow$ small
12:    **end if**
13:    $\delta_{\text{prev}} \leftarrow \delta_{\text{cur}}$
14:    $x_0 \leftarrow x_1$
15:    $k \leftarrow k + 1$
16: **end while**
17: $x^\star \leftarrow x_0$

---

