# OpenReview forum: "Two-Scale Latent Dynamics for Recurrent-Depth Transformers"
_NeurIPS.cc/2025/Workshop/UniReps — UniReps2025_

### Official Review · Reviewer_wijD · 2025-09-03
**Second-Order Early-Exit**

**Confidence:** 3

**Review:**

## Summary

* The paper proposes an *acceleration test* to improve the stopping rule for a recurrent-depth Transformer based on Ref. [5] (Huginn), which previously used two approaches: a per-token *velocity test* ("step norm" in the paper) or a *KL test* (slower but better). Experiments with GPT-2 on FineWeb (perplexity/cross-entropy) show that acceleration achieves performance comparable to KL while substantially improving the inference speed.
* The theoretical motivation is an analysis of the angles between consecutive iterations, which converge to a fixed, nonzero value.
* Evidence is visualized via a 2D PCA trajectory of a GPT-2 with Prelude (Layers 1–3) → Repeat (4) → Repeat (5/6) → Repeat (7) → Coda (8–12), which clusters around the three recurrent stages. The smaller-scale evolutions are indeed spiral.

## Strengths

* Scaling inference-time compute via recurrent Transformers (e.g., Universal Transformers, Ref. [2]) has significant practical impact. Avoiding repetitive decoding (I find it particularly interesting to reuse the decoding matrix) in the KL criterion is practically useful.
* The idea is well motivated: an autonomous system exhibits two types of evolution—contractive/diffusive (scale-dependent, related to the step norm) and unitary/conservative (e.g., a harmonic oscillator or the spiral/loop geometry discussed in the paper). These correspond to the real and complex eigenvalues of the evolution generator—the UNIversal REPresentation of ODE limits is a circle rather than a single point.

## Weaknesses

I do not find significant weaknesses in the paper and support an acceptance, good luck. Minor weaknesses are left as comments below.

## Discussions

* I do not see why the authors emphasize *two-scale* dynamics instead of *spiral* evolution. Using three rather than one loop sequence (as in Ref. [5]) and showing a clear separation of clustered stages makes for an interesting discussion of local geometry, but it seems optional as the primary motivation. If the authors are curious about the potential benefits of varying the number, position, or length of loop stages instead of fixing them arbitrarily at (1, 2, 1), they are encouraged to validate this (perhaps in future work) or explain it in an update.
* It took me some time to realize that the KL metric is not computed in latent space (as with comparing Gaussian priors in a VAE) but uses the *same* final layer to decode (the UNIversal residual REPresentation) into the vocabulary probability space—where softmax compresses many directions of the feature space except the one affecting the output token, so spirals may not matter. It might help readers to understand more quickly if the authors clarify KL is slower due to the O(dV) decoding multiplication (which dominates the O(V) softmax).

**Score:**

4

**Topic Fit:**

2

---

### Official Review · Reviewer_8Hro · 2025-09-13
**Unclear how this work is related to workshop**

**Confidence:** 2

**Review:**

The work appears to be a incremental extension of Geiping et al. [5], from which it borrows the core model, training setup, and baseline methods. The central contribution is an observational characterization of update steps and a new "acceleration"-based early-exit rule, which seems to me a small second-order modification. The work would be stronger if it more clearly articulated a central contribution beyond a refinement of learning updates.

Moreover, I am not sure how closely this is related to the topic of the workshop, as there is little I can see related to characterizing or providing principled reasoning behind the representations of artificial neural systems. The figures presented seem to be course characterization of learning dynamics. I am not sure how to draw any principled claims about neural representations from these results.

**Score:**

1

**Topic Fit:**

1

---

### Official Review · Reviewer_94Qk · 2025-09-15
**Two-Scale Latent Dynamics & “Acceleration” Early-Exit for Recurrent-Depth Transformers**

**Confidence:** 4

**Review:**

**Summary**
This extended abstract argues - using iterate-space diagnostics - that recurrent-depth Transformers exhibit a two-scale geometry: small, increasingly orthogonal refinements within looped blocks and coarser drift across consecutive blocks. Leveraging this, the authors propose a simple second-order early-exit rule based on the change between consecutive latent updates (“acceleration”), claiming better latency–quality trade-offs than step-norm and KL exits. The work is clearly written, empirically grounded, and well aligned with UniReps’ learning dynamics theme, though it would benefit from stronger quantitative evidence, richer representation-analysis (e.g., CKA/RSA), and broader validation.

**Quality**
Strong extended-abstract: clear writing, crisp problem framing, and a simple, plausible contribution (acceleration-based early exit) grounded in intuitive iterate-space diagnostics. The two-scale latent-dynamics narrative is insightful and of interest to UniReps, and the proposed criterion is cheap, decoding-free, and practically useful. However, evidence is narrow (single GPT-2–scale setup), quantitative reporting is light (few hard numbers/variances), and the connection to representation-analysis tools (CKA/RSA, principal angles, curvature) is underdeveloped; baseline coverage beyond step-norm/KL is also limited. Overall, it’s promising and well presented for a non-archival track, but it needs broader validation and richer alignment analysis to be fully convincing.

**Clarity**
Writing is clear and concise; figures (as described) seem informative (PCA overview with inset loop; norm/angle plots by group; latency/quality vs. threshold). The limits section is candid and appreciated. Some references are year-fresh and relevant to dynamic compute and latent reasoning.

**Originality**
The two-scale iterate-geometry framing of recurrent depth is a neat conceptual contribution, and the specific second-order exit rule (acceleration + two-hit + normalization) is a simple, arguably novel twist on early exit in latent space.

**Significance**
If the effect generalizes, practitioners can adopt acceleration-based halting to save TTC without decoding overhead. For UniReps, the measured evolution of step angles/norms speaks directly to learning dynamics and representational geometry.

**Pros**
- Clear two-scale geometric narrative with simple, reusable diagnostics.
- Practical acceleration-based halting with decoding-free cost and good qualitative trade-offs.
- Well scoped to learning dynamics; writing is clean and limitations are explicit.

**Cons**
- Limited scope (single model/setup), primarily observational evidence.
- Missing deeper representation-alignment analyses (CKA/RSA, principal angles across subspaces).
- Baseline coverage could be broader; quantitative tables and reproducibility details are light.

*Cons are not critical as the work targets extended abstract track and authors are aware of those limitations*

**Score:**

4

**Topic Fit:**

3

---

### Official Review · Reviewer_xQ6q · 2025-09-15

**Confidence:** 4

**Review:**

**Summary**

This paper studies the latent dynamics of recurrent-depth transformers, focusing on the geometry of model embeddings when blocks are iterated at test time. The authors propose a two-scale view: within a looped block updates act as small refinements that shrink and become increasingly orthogonal, forming spiral-like trajectories, while across consecutive blocks, representations exhibit larger-scale drift. To characterize these dynamics, they analyze step norms, consecutive-step angles, and visualize trajectories using PCA. Based on these insights, the paper introduces a second-order early-exit rule that halts inference when consecutive loop updates stop changing.

**Strengths**

1. The paper provides a clear and intuitive conceptual framework of recurrent-depth transformers, distinguishing between small-scale loop refinements and larger cross-block drifts.
2. Motivated by the analysis of embeddings within recurrent blocks, the paper introduces a simple yet practical inference acceleration rule that outperforms the tested baselines.

**Weaknesses**

1. Some aspects of the observed behaviour have been noted in prior work [1] (e.g., Figure 12), where similar dynamics were demonstrated but not explored in depth. While this paper extends the analysis by quantifying step norms and examining behaviour across multiple recurrent blocks — which represents a strong and novel contribution — it should explicitly acknowledge the connection to previous observations.
2. Beyond PCA visualizations, the paper provides no evidences for the larger-scale changes across recurrent blocks. Including additional metrics, such as step norms or alternative quantitative measures, would strengthen the claim and provide a more robust argument for cross-block dynamics.
3. The authors claim that the proposed early-exit rule achieves “a better latency–quality Pareto than step-norm and KL-based baselines” (line 43); however, no explicit Pareto plot is included. Including such a plot would make the comparison clearer. Especially since the cross-entropy and perplexity subfigures (ii) and (iii) are closely related and differ only by axis scale.

**Minor comments and questions**

* A clearer separation between training dynamics and inference-time behavior within text could improve the readability and structure of the paper.
* I believe the Figure 4 better illustrates the claim stated e.g in lines 27,39 or 63 than Figure 3b.
* Could injecting noise separately at each recurrent block contribute to the observed larger drift between blocks?
* What is the intuition behind using a two-layer recurrent block in some regions versus a single-layer recurrent block in others?

**References**

[1] Geiping, Jonas, et al. "Scaling up test-time compute with latent reasoning: A recurrent depth approach." arXiv preprint arXiv:2502.05171 (2025).

**Score:**

4

**Topic Fit:**

2